# Predicting CO$_2$ Emission Footprint Using AI through Machine Learning

**Yang Meng [1],\* and Hossain Noman [2]**

[1]  School of Mechanical Electronic Information Engineering, China University of Mining and Technology, 11 Xueyuan Road, Haidian District, Beijing 100083, China

[2]  Department of Computer Science & Engineering, University of Liberal Arts Bangladesh, 688 Beribadh Road, Mohammadpur, Dhaka 1207, Bangladesh

\*  Correspondence: m.yang@cumtb.edu.cn

**Abstract:** Adequate CO$_2$ is essential for vegetation, but industrial chimneys and land, space and oceanic vehicles exert tons of excessive CO$_2$ and are mostly responsible for the greenhouse effect, global warming and climate change. Due to COVID-19, CO$_2$ emission was in 2020 at its lowest level compared to prior decades. However, it is unknown how long it will take to reduce CO$_2$ emission to a tolerable point. Furthermore, it is also unknown to what extent it can increase or change in the future. Accurate forecasting of CO$_2$ emissions has real significance for choosing the optimum ways of reducing CO$_2$ emissions. Although some existing models have noticeable CO$_2$ emission forecasting accuracy, the models implemented in this work have more efficacy in prediction due to incorporating COVID-19's effect on CO$_2$ emission. This paper implements four prediction models using SARIMA (SARIMAX) based on ARIMA. The four models are based on the time period of the surge of the COVID-19 pandemic. The main objective of this work is to compare these four models to suggest an effective model to predict the total CO$_2$ emissions for the future. The study forecasts global total CO$_2$ emission from 2022 to 2027 for near future prediction, 2022 to 2054 for future prediction and 2022 to 2072 for far future prediction. Among the various error measures, mean absolute percentage error (MAPE) is chosen for accuracy comparison. The calculation yields different accuracy for the four SARIMAX models. The MAPEs for the four methods are: pre-COV (MAPE: 0.32), start-COV (MAPE: 0.28), trans-COV (MAPE: 0.19), post-COV (MAPE: 0.09). The MAPE value is relatively low for post-COV (MAPE: 0.09). Hence, it can be inferred that post-COV are suitable models to forecast the global total CO$_2$ emission for the future. The post-COV predictions for the global total CO$_2$ emission for the years 2022 to 2027 are: 36,218.59, 36,733.69, 37,238.29, 37,260.88, 37,674.01 and 37,921.47 million tons (MT). This study successfully predicts CO$_2$ emission either for the COVID-19 period or the post-COVID-19 normal periods. The Machine Learning (ML) method used in this study has shown good agreement with the IPCC model in predicting the past emissions, the current emissions due to COVID-19 and the emissions of the upcoming future. These prediction results can be an asset for the decision support system to develop a suitable policy for global CO$_2$ emission reduction. For future research, a number of other external influence variables responsible for CO$_2$ emission can be added for finer forecasts. This research is an original work in predicting COVID-19-affected CO$_2$ emission using AI through the ML methodology.

**Keywords:** Artificial Intelligence; machine learning; CO$_2$ emission; global warming; atmosphere monitoring; atmosphere maintenance

## 1. Introduction

A certain amount of CO$_2$ is essential for the environment we live in. Excessive CO$_2$ emissions have some impact on the environment. Industrialization and other human activities are constantly putting a large amount of CO$_2$ into the atmosphere. Prior to the COVID-19 pandemic, the world had experienced the highest amount of CO$_2$ emission

ever seen. During and transmission (trans) time of COVID-19, the emission of $CO_2$ has descended to 34.4 million tons (MT), which is lower than the previous peak (36.1 MT) [1]. There are numerous works that estimate $CO_2$ emission before the pandemic but there is no suitable work showing how $CO_2$ emissions will behave in the trans- and post-COVID-19 era, because most of the recent works either use data from before the pandemic, such as [2] (up to 2018), [3] (up to 2018), [4] (up to 2015), or they use data from during pandemic but with a local scope, such as [2] for India, [3] for Turkey, [4] for the UK, [5] for China and [6] for indoor environments, or they use different approaches for only near future (2 years) forecasting [7]. This research focuses on developing a Machine Learning (ML)-based Artificial Intelligence (AI) model to predict $CO_2$ emission in the near and far future considering the reduced $CO_2$ emissions due to the lockdowns for the COVID-19 pandemic.

### 1.1. Global $CO_2$ Emission Crisis

It is well known that $CO_2$ emission is a major issue for global warming due to the greenhouse effect [8]. Although there is controversy over whether $CO_2$ is responsible for global warming or not [8], despite this controversy there is strong consensus, e.g., [9,10], that $CO_2$ emission is mainly responsible for global warming. As a result, assessment as well as forecasting of the $CO_2$ emission footprint are important for various aspects: Firstly, to assess $CO_2$ emission to identify major contributors to global warming, since $CO_2$ emission is considered as the main contributor to global warming [11] and climate change [12]. Secondly, to understand the $CO_2$ emission footprint to develop a policy to fight against it. Thirdly, to compensate for environmental or financial losses incurred by $CO_2$ emission. Fourthly, to assess the rational effect of $CO_2$ emission on GDP reduction [13], stock market casualty [14], new or old diseases upheaval [15], air quality disruption [16] and the effect on building a greener and cleaner smart city. Most importantly, forecasting of $CO_2$ emission is essential to measure and defeat irreversible climate change [12].

### 1.2. Literature Review

To date, there exist some works that have modeled the global $CO_2$ emission footprint, including the COVID-19 transmission period, such as [7]. Most of the existing works have either a partial to local context such as [17] in China, [18] in China, [19,20] in wheat fields, [21] in Iran, [22] in the Middle East, or the modeling parameters and methodology are not appropriate for global $CO_2$ emission prediction, such as [3] for Indian paddy fields, [2] for the Turkish transportation sector.

The strengths and limitations of existing local works are presented chronologically below. Local $CO_2$ emission was forecasted for the case of the Iranian domain using ML and artificial neural network-based modeling in [21]. The $CO_2$ emissions from fossil fuel and cement production are presented in [23]. $CO_2$ driver and emission forecasting was conducted based on a local county named Changxing in China in [17]. Moreover, ref. [24] analyzes and forecasts the emissions from $CO_2$ using the dataset of the years 1995 to 2018 from the Indian region. $CO_2$ emissions in the Arabian region are presented in [22]. The synergistic effect of $CO_2$ emissions on PM2.5 emission reduction in the Chinese region is presented in [25]. Moreover, ref. [26] provides a decent overview of $CO_2$ emission and its related issues but is lacking concerning building a $CO_2$ emission model for a global case. To date, the most accurate forecasting of $CO_2$ can be found with the model developed by the IPCC [27]. It provides predictions of $CO_2$ emissions such as 398 ppm (2019), 400 ppm (2020), 402 ppm (2021) and 405 ppm (2022). Here, ppm means parts per million, a unit for $CO_2$ emission measurement. The forecasting results with the IPCC model are good enough for the non-pandemic years, but the model shows degrading behavior in predicting emission values for the period of pandemic surge. As a result, a more inclusive model needs to be introduced.

A number of modeling approaches have been tried by various authors from different perspectives to forecast $CO_2$ emissions. Notably, ref. [28] provides insight into the $CO_2$ emission prediction model using ML. $CO_2$ emissions and environmental protection issues

have brought pressure from the international community during China's economic development [29] era. A novel hybrid model using combined principal component analysis (PCA) was build based on the data from 1978 to 2014 for China in [19]. Additionally, ref. [30] bring out the trends in $CO_2$ emission from fossil fuels in Zambia from 1964 to 2016. A prediction model for $CO_2$ emissions based on multiple linear regression analysis in the Chinese context was studied in [31]. Furthermore, two models have been developed for simulating $CO_2$ emissions from wheat farms [20] in New Zealand. Moreover, the SVM model was proposed to predict expenditure of carbon ($CO_2$) emission in [32]. A data mining approach to find $CO_2$ emission from vehicular data is presented by [33]. The back-propagation artificial neural networks (ANN) model was presented to predict expenditure of carbon ($CO_2$) emission in [34]. A quantitative study to evaluate the effect of $CO_2$ on temperature change in five regions was presented in [35]. It finds that $CO_2$ is responsible for 50.2% of the global temperature rise during 1990–2010 [35]. A similar finding was also true for the next decade (2010–2019) until the COVID-19 pandemic surges across the world, as seen in Figure 1 [27]. The findings of the related literature review are summarized in Table 1.

**Table 1.** Summery of Literature Review on $CO_2$ Emission Forecast.

| Article | Method | Accuracy | Context | Findings |
|---------|--------|----------|---------|----------|
| [28] | ML(Regression) | RMSE 25.57% | Global | Achieve low RMSE |
| [29] | SVM-ELM | RMSE 12.34% | China | $CO_2$ prediction up to 2030 |
| [19] | PCA | RMSE 0.3% | China | $CO_2$ prediction up to 2014 |
| [30] | WEKA | - | Zambia | Forecast $CO_2$ emission |
| [31] | Regression | Error 2.5% | China | Accuracy in prediction |
| [20] | ANN | MSE 11% | New Zealand | Better accuracy |
| [32] | SVM | MSE 0.04% | Indonesia | Effective decision making |
| [33] | Data Mining | - | Vehicular data | Accuracy in prediction |
| [34] | ANN | RMSE 5.5% | Sugar Industry | Accuracy in prediction |
| [35] | Quantitative | - | Global | 50.2% increase of temperature |
| [27] | $C^4$MIP model | MSE (10–20)% | Global | Accuracy in prediction |

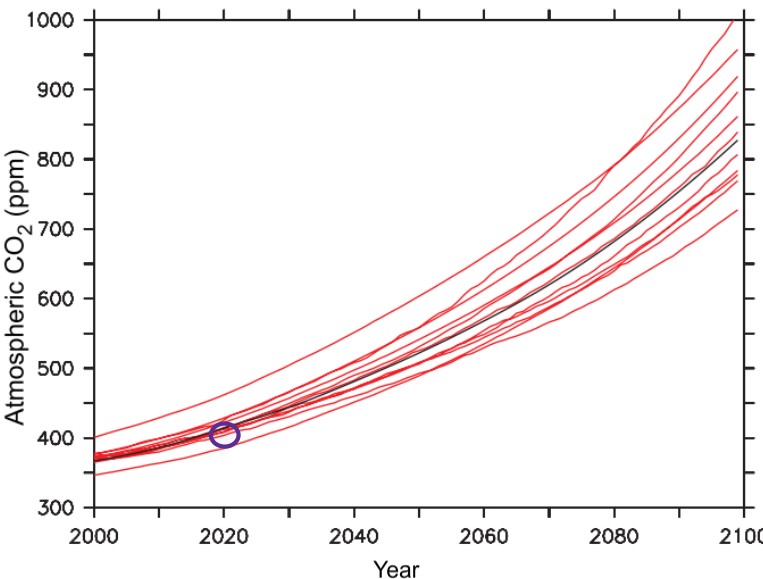

**Figure 1.** $CO_2$ Emission Forecasts by IPCC Model [27].

Previous studies had limitations in qualifying or quantifying measurements of $CO_2$ emissions during and after the COVID-19 era. A more accurate $CO_2$ emission model is

required to estimate the temperature rise for the next decade, 2020–2030, as well as later decades such as 2030–2040, 2040–2050 etc.

### 1.3. Research Objective

The objective of this work is to model accurate $CO_2$ emission behavior for the past, present and near future. This study can be considered the latest nexus of previous work, as most of the previous works did not include the changes in $CO_2$ emissions during the COVID-19 pandemic. Although prior works revealed some similar modeling approaches, this study is completely different in scope, accuracy, context and forecasting. Moreover, this study contributes to the literature in a few notable points. First, contrary to previous attempts, this study further uses current data with historical data to provide recent trends for the data modeling process. As a result, the final model involves concurrent reduced $CO_2$ emissions (to 5.2%) [1] during the COVID-19 pandemic. This paper divided COVID-19's effect on $CO_2$ emissions into four periods (prior (pre), start, transmission (trans) and post) and it also prepared four respective datasets. Second, this study uses extant time series-based ML models to develop the $CO_2$ emission forecasting model, but there is a difference concerning the optimum model selection process. The model selection process is different in the sense that it obtains the best model and related parameters. Based on the selected parameters, the developed model become accurate (less error prone). As a result, near and far future forecasting become accurate as compared to real $CO_2$ emissions of that time. Initially, the existing time series-based models were chosen; the authors developed the ML model optimization algorithm that selects the best model for each $CO_2$ dataset to obtain the best possible forecast. Lastly, this study forecasts the $CO_2$ emission footprint for the near future, e.g., 6 years (from 2022 to 2027) or 32 years (from 2022 to 2054), and the far future, e.g., 50 years (from 2022 to 2072), as an example.

## 2. Materials and Methods

This paper used all available annual data for global $CO_2$ emissions. It used self-developed algorithms to clean the data and select the best ML models from the data. It then used the selected algorithm to develop forecasting models for the prediction of $CO_2$ emission behavior. Afterward, validation and comparison were performed to evaluate our forecasting models and model results. The complete modeling procedure is given in the following subsections.

### 2.1. Data and Processing

This paper primarily used global annual $CO_2$ emission data from 1751 to 2018. The primary data were retrieved from this repository [36]. These data were then cleaned, engineered and processed. As a result, four sets of data were prepared depending on the occurrence of the COVID-19 pandemic (prior (pre), start, transmission (trans) and post). The data up to December 2018 are called pre-COVID-19 data, the data up to December 2019 are named start-COVID-19 data, the data up to December 2020–2021 are called the trans-COVID-19 dataset and periods after that (e.g., 2022–2023) are called the post-COVID-19 dataset. The time periods trans-COVID-19 and post-COVID-19 are relative periods. If the COVID-19 pandemic globally disappeared in 2021, then post-COVID-19 periods could include 2022; otherwise, we will consider later years for the post-COVID-19 periods. To add phenomenal reality to the data, the time period of 2020 to 2021 is regarded as trans-COVID-19 (since COVID-19 was severely present all over the world). The emissions datasets are shown in Table 2.

**Table 2.** $CO_2$ Emission Datasets.

| Dataset Name | Time Period |
|:---:|:---:|
| pre-COVID-19 | 1751–2018 |
| start-COVID-19 | 1751–2019 |
| trans-COVID-19 | 1751–2021 |
| post-COVID-19 | 1751–2023 |

### 2.2. Data Preparation and Augmentation

Once the data were available at hand, the next step was to understand the data. The data taken from the source were in the range of 1750 to 2018. Since the COVID-19 pandemic has surged over the globe since 2019, its effect on $CO_2$ emissions must be included in the forecasting model. For this reason, a number of data points have been included in the recorded data. Yearly data for 2019–2023 are included in the recorded dataset.

### 2.3. Feature Selection

The dependent variable in this study was the amount of total $CO_2$ emission measured in MT, while the independent variable is the year. In this study, data were divided into training data and testing data; the train–test split was maintained at an 8:2 ratio. Training data were used in the $CO_2$ emission estimation process of the model while testing data were used to determine the accuracy of the prediction of $CO_2$ emissions. Data from December 1751 to December 1994 were considered as training data and data from December 1995 to December 2018 were used as testing data for the pre-COVID-19 model. A similar train–test split was also effected for the three remaining datasets by stepping one year forward for each of them.

To augment data for the years from 2020 to 2023, the authors used the data from the source given here [1]. The respective changes in data for the other years were determined by considering the similar rates that were found in 2020 to 2021. The data trend is visualized in Figure 2. In Figure 1, actual and augmented data are clearly visible. Figure 2a,b present data from before the COVID-19 period. From Figure 2c, it can be seen that $CO_2$ emissions decreased radically during the pandemic. Similar behavior can also be observed in Figure 2d for the remaining augmented cases such as 2021, 2022, etc. This data augmentation process takes advantage of developing the actual $CO_2$ emission model to trace future emission behavior. Moreover, the data augmentation for near-term $CO_2$ emissions will help to reduce modeling errors; thus, it helps in building real and suitable models. The best performing model can help in creating robust policies for the future to fight against $CO_2$ emission problems across the world. For example, if one chooses to build a model based on the data available in Figure 2a, the model could predict wrong values. The model may not explain current or future emission behavior well during the changing environment of the pandemic. Moreover, there is a large chance that a forecasting model only based on data from 1751 to 2018 could be inaccurate. Seasonal data during COVID-19 should be included. The varied data values for different datasets are presented in Table 3.

**Table 3.** Augmented Values of $CO_2$ Emissions in Various Datasets.

| Dataset Name | Year | Values |
|:---:|:---:|:---:|
| pre-COVID-19 | 2018 | 36,572.75 |
| start-COVID-19 | 2019 | 36,441.45 |
| trans-COVID-19 | 2020 | 34,226.18 |
| trans-COVID-19 | 2021 | 36,190.82 |
| post-COVID-19 | 2022 | 33,900.52 |
| post-COVID-19 | 2023 | 34,164.68 |

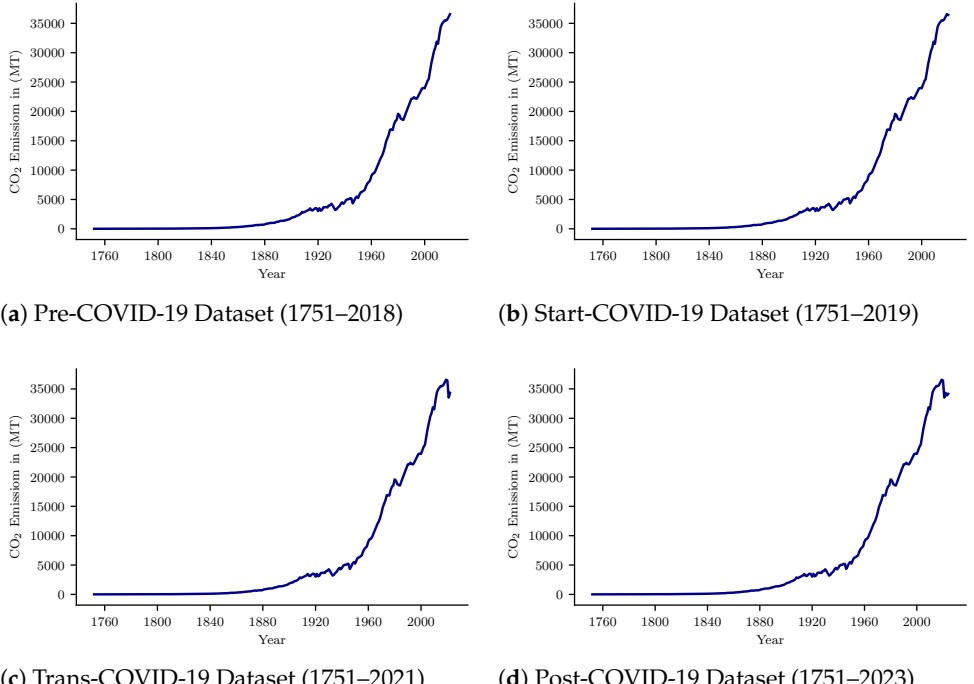

(**a**) Pre-COVID-19 Dataset (1751–2018)

(**b**) Start-COVID-19 Dataset (1751–2019)

(**c**) Trans-COVID-19 Dataset (1751–2021)

(**d**) Post-COVID-19 Dataset (1751–2023)

**Figure 2.** $CO_2$ Emission Dataset Visualization.

### 2.4. Data Modeling

To model $CO_2$ emissions, the authors used a time series-based ML technique named Autoregressive Integrated Moving Average (ARIMA) as well as Seasonal Autoregressive Integrated Moving Average (SARIMA). SARIMA is similar to ARIMA but seasonality is added to it. These two algorithms are regarded as the robust model and they are capable of presenting both stationary and non-stationary time series data. To forecast time series, three conditions need to be checked: (a) tentative identification, (b) parameter estimation and (c) diagnostic checking. Auto-regressive models are adroit in modeling different kinds of time series; (a) auto-regressive (AR), (b) moving average (MA), (c) auto-regressive moving average (ARMA) and (d) ARIMA. The base for the ARIMA model is the Box–Jenkin method [37]. ARIMA is written as ARIMA (p,d,q) where the seasonal parameter is absent and SARIMA is written as SARIMA (p, d, q) (P, D, Q)$^S$ where S is the seasonal parameter. During the ARIMA model optimization process $S = 19$ was found to be the best seasonality parameter, thus the ARIMA model turns into a SARIMA model and is presented as SARIMAX.

The SARIMA model can be written as:

$$\varphi_p(B)\Phi_p(B)B^S(1-B)^d(1-B^S)^D Y_t = \theta_q(B)\Theta_Q(B^S)\varepsilon_t \tag{1}$$

where:

$$\varphi_p B = 1 - \varphi_1 B - \varphi_2 B^2 - \ldots - \varphi_p B^p$$

$$\Phi_p(B)B^S = 1 - \Phi_1 B^S - \Phi_2 B^{2S} - \ldots - \Phi_p B^{pS}$$

$$\theta_q(B) = 1 - \theta_1 B - \theta_2 B^2 - \ldots - \theta_q B^q$$

$$\Theta_Q(B^S) = 1 - \Theta_1 B^S - \Theta_2 B^{2S} - \ldots - \Theta_Q B^{QS}$$

In the equations above, $t = 1, 2, 3 \ldots N$; $N$ is the number of observations up to time t; B is the backshift operator defined by $B^\alpha W_t$; $\varphi_p(B)$ is called a regular (non-seasonal) autoregressive operator of order p; $\varphi_p(B^S)$ is a seasonal autoregressive operator of order p; $\theta_q(B)$ is a regular moving average operator of order q; $\Theta_Q(B^S)$ is a seasonal moving

average operator of order Q; $\varepsilon_t$ is identically and independently distributed as normal random variables with mean zero, variance $\alpha^2$ and $cov(\varepsilon_t), \varepsilon_{t-k} = 0, \forall k \neq 0$; p is the auto-regressive term; q is the moving average order; P is the seasonal period length of the model, S, of the auto-regressive term; Q represents the seasonal period length of the model, S, of the moving average order; D represents the order of seasonal differencing; d represents the order of ordinary differencing [38].

While fitting data to a SARIMA model, the values of d and D are estimated initially; this gives good results during seasonality issues. The remaining values of p, q and Q need to be chosen by the auto-correlation function (ACF) and the partial auto-correlation function (PACF). AFC and PACF were automatically calculated by the program developed for data modeling. To control overfilling in the models, hold-outs (test–train split), feature selection and data augmentation techniques were used.

To evaluate the model, we use some prediction metrics, namely mean absolute percentage error (MAPE) [39], mean squared error (MSE) [40], root mean squared error (RMSE) [39] and mean absolute deviation (MAD) [39]. For simplicity and integrity, MAPE scores were finally presented in this paper for model accuracy comparison.

To build different models, the ARIMA algorithm was repeatedly executed using the author-developed optimization algorithm. After checking efficiency issues, the most efficient model was used. Models that were found to be efficient with regard to this work were as follows; for the pre-COVID-19 period, the ARIMA (2,1,2)(0,1,1) [19] (SARIMAX(2, 1, 2)x(0, 1, 1, 19)) model with ACF = 0.88 and MAPE = 0.32; for the start-COVID-19 period, the ARIMA(1,1,2)(0,1,1) [19] (SARIMAX(1, 1, 2)x(0, 1, 1, 19)) model with ACF 0.93 and MAPE = 0.28; for the trans-COVID-19 period, the ARIMA(0,2,1)(1,1,1) [19] (SARIMAX(0, 2, 1)x(1, 1, 1, 19)) model with ACF 0.90 and MAPE = 0.19; and for the post-COVID-19 period, the ARIMA(0,2,1)(1,1,1) [19] (SARIMAX(0, 2, 1)x(1, 1, 1, 19)) model with ACF 0.88 and MAPE = 0.09.

Here, the authors introduced three metrics, namely $\Delta CS$, $\Delta CT$ and $\Delta PC$, to calculate the forecasting value difference between different models to show the error propagation among different models. These metrics can show us the forecasting error between different models. This is the difference between the pre-COV (pre-COVID-19) model predicted values and the respective models during the COVID-19 surge (e.g., start-COVID-19, trans-COVID-19 and post-COVID-19). This means that if one chooses to forecast actual $CO_2$ during and after the COVID-19 period, one needs to select any model other than pre-COVID-19; otherwise, a substantial error will spread in the forecasting value over time.

$$MAPE = \frac{\sum_{t=1}^{n} |(y_t - \hat{y}_t)/y_t|}{n} 100 \tag{2}$$

$$MSE = \frac{\sum_{t=1}^{n} (y_t - \hat{y}_t)^2}{n} \tag{3}$$

$$RMSE = \sqrt{\frac{\sum_{t=1}^{n} |y_t - \hat{y}_t|}{n}} \tag{4}$$

$$MAD = \frac{\sum_{t=1}^{n} |y_t - \hat{y}_t|}{n} \tag{5}$$

$$\Delta CS = preCov_{pred} - startCov_{pred} \tag{6}$$

$$\Delta CT = preCov_{pred} - transCov_{pred} \tag{7}$$

$$\Delta PC = preCov_{pred} - postCov_{pred} \tag{8}$$

After the model is successfully developed, it is time to create visual representations of the modeling outcomes. Figures 3 and 4 present the outcomes of the models that were built beforehand. Figure 3 presents the internal forecasting results related to the time period (either for 2018, 2019, 2020 or 2021) of the datasets and Figure 4 presents the external or future (time periods beyond the datasets, that is, 2022, 2023, etc.) forecasting behavior.

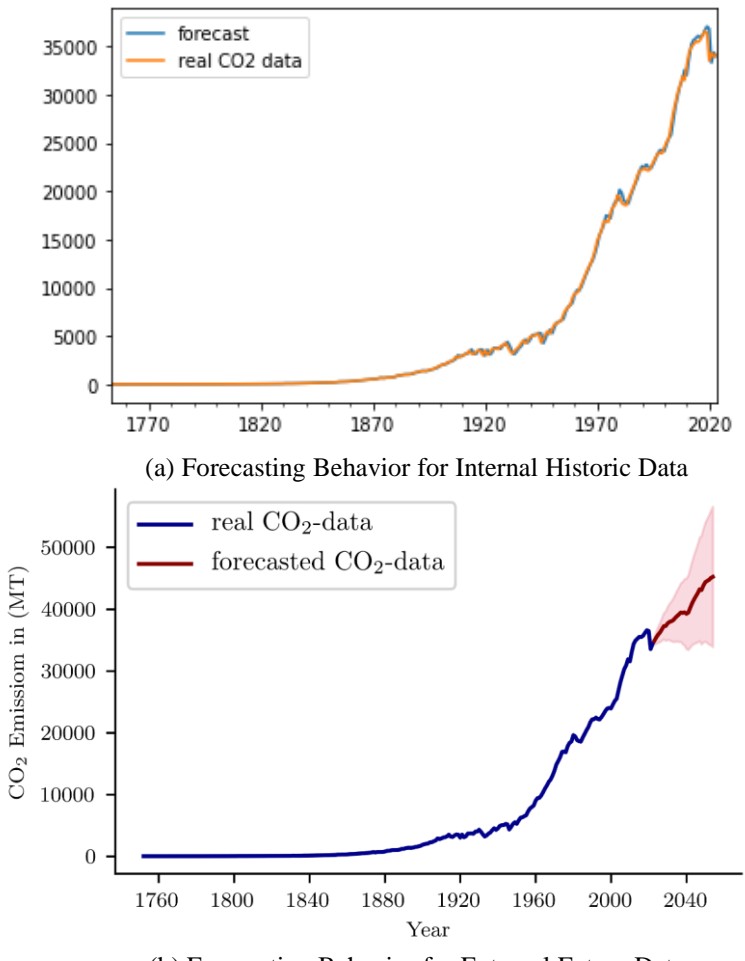

(a) Forecasting Behavior for Internal Historic Data

(b) Forecasting Behavior for External Future Data

**Figure 3.** CO$_2$ Emission Forecasting Behavior for Internal and External Time.

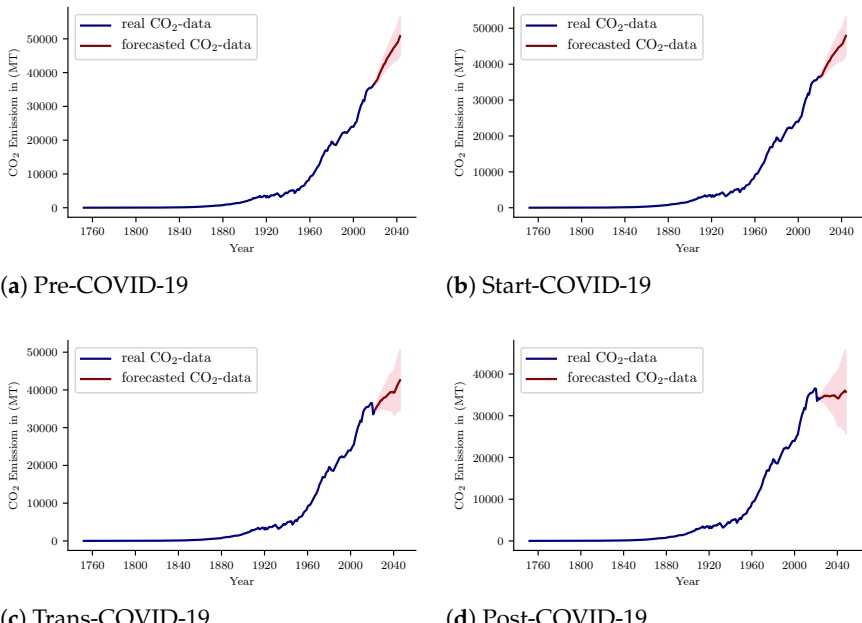

(**a**) Pre-COVID-19

(**b**) Start-COVID-19

(**c**) Trans-COVID-19

(**d**) Post-COVID-19

**Figure 4.** CO$_2$ Emission Forecasting Behavior for Different Cases of Data.

During data modeling, four things can happen—one can build a model: (1) based on the base data (pre-COVID-19 data as shown in Figure 2a) before the COVID-19 pandemic; (2) based on the data (start-COVID-19 data as shown in Figure 2b) when the COVID-19 pandemic starts surging; (3) based on the data (trans-COVID-19 data as shown in Figure 2c) while the COVID-19 pandemic is spreading globally; (4) based on the data (post-COVID-19 data as shown in Figure 2d) after the COVID-19 pandemic is over.

If one intended to build a $CO_2$ emission model with option (1), forecasting results will probably not represent the real situation concerning emissions observed due to the COVID-19 pandemic. Option (2) would not be the justified option for the same reason, with regard to the period of the COVID-19 pandemic just beginning to spread over China. Option (3) would be the viable option for building a model to forecast $CO_2$ emissions because in this time period the COVID-19 pandemic spreads over the world and massive lockdown processes have already shut down a huge number of $CO_2$ emission sources acriss the world. Option (4) would be a supplementary one if COVID-19 finishes its surge over the globe in this period. The time span of 2022 to 2023 can be considered the period when the COVID-19 pandemic will finish its surge; if it is not, then this period can be considered as part of the extended transmission period. As seen from the global situation, the COVID-19 surge ended in 2021. So, option (4) can only be a post-COVID-19 situation.

If one wants to forecast the exact behavior of $CO_2$ emissions well, these authors suggest building all four models (at least 3 models from 2 to 4), so that exact emission behavior can be covered. No single model can forecast the exact $CO_2$ emissions well. These authors in the end chose a model the reflects the $CO_2$ emissions in the near or far future.

*2.5. Model Validation*

To validate the models presented in this paper, Figure 3 is sufficient for the evidence. Figure 3 shows the forecasting behavior of emissions for the current (1751–2021) and future (2022 and beyond) years. The pink shadow in Figure 3 is the confidence interval (the upper and lower bound of forecast). Furthermore, a number of performance parameters are presented here to better understand the forecasting values. Table 4 presents the modeling error and accuracy parameters as found during the model development.

**Table 4.** Performance Parameters for $CO_2$ Emission Models.

| Dataset | Model | MAPE | Accuracy |
|---|---|---|---|
| pre-COVID-19 | ARIMA(2,1,2)(0,1,1) [19] | 0.32 | Reasonable |
| start-COVID-19 | ARIMA(1,1,2)(0,1,1) [19] | 0.28 | Reasonably Better |
| trans-COVID-19 | ARIMA(0,2,1)(1,1,1) [19] | 0.19 | Accurate |
| post-COVID-19 | ARIMA(0,2,1)(1,1,1) [19] | 0.09 | Highly Accurate |

As seen from Table 5, the error scores for the models are 32%, 28%, 19% and 9% for the respective models. In accordance with Table 4 [41], the accuracy intensities for the respective models are named Reasonable, Reasonably Better, Accurate and Highly Accurate.

**Table 5.** Interpretation of Typical MAPE Value.

| MAPE | Interpretation |
|---|---|
| >50% | Inaccurate Forecasting |
| 20–50% | Reasonably Forecasting |
| 10–20% | Accurate Forecasting |
| 10% | Highly Accurate Forecasting |

Hence, the best models are the models that use data from during the COVID-19 pandemic surge. The outcomes of the models exactly resemble the reality of $CO_2$ emissions across the globe. As the COVID-19 pandemic reaches its mild stage across the globe and lockdowns end, this situation can be treated as the post-COVID-19 period. As a result, to be

in line with the real world situation, the post-COVID-19 model actually reflects the current situation concerning $CO_2$ emissions. We predicted some values of $CO_2$ emissions for a few years and compared them with real emission data [42] as well as a benchmark IPCC model [27]. The comparison results shows the model performs well against real world and benchmark IPCC models. All the results are measured in ppp and giga tons (GT). They are presented in Table 6.

**Table 6.** Validation of $CO_2$ Emission Forecast measured in (ppm-GT).

| Year | Current [42] | IPCC [27] | Our Model | Authorś Remarks |
|------|------|------|------|------|
| 2022 | 415.76 [43]–Yet to get | 405–36.07 | 409–36.45 | Accurate Forecast |
| 2021 | 413.79 [43]–36.4 | 402–35.83 | 406–36.18 | Highly Accurate Forecast |
| 2020 | 412.44 [43]–34.8 | 400–35.65 | 395–35.20 | Accurate Forecast |
| 2019 | 410.07 [43]–36.7 | 398–35.57 | 408.85–36.44 | Accurate Forecast |

As seen from Table 6, forecasting models are justified and accurate enough to represent real $CO_2$ emission behavior for the current and near future. It is inferred that far future predictions would be justified too. For purposes of further forecasting, in the end the most accurate model (post-COVID-19 model) was chosen to present the different forecasting scenarios.

## 3. Results

A number of interesting and insightful results were found upon forecasting near–far future $CO_2$ emissions using the accurate models. Near future results are used to validate the model. Afterwards, far future $CO_2$ emissions are predicted. Results are presented in the following sections.

### 3.1. Near Future Emission Forecast

The selected Post-COVID-19 model was used to forecast the near future $CO_2$ emission values. The model yields some empirical forecasting for the years selected. The forecasting results are shown in Table 7. These near future forecasting values will be the supportive evidence for the far future forecasting by the model.

**Table 7.** Near Future $CO_2$ Emission Forecast (MT).

| Year | $CO_2$ Emission |
|------|------|
| 2022 | 36,218.59 |
| 2023 | 36,733.69 |
| 2024 | 37,238.29 |
| 2025 | 37,260.88 |
| 2026 | 37,674.01 |
| 2027 | 37,921.47 |

### 3.2. Increasing Progression in $CO_2$ Emission over Time

Forecast models show an increasing growth rate of $CO_2$ emissions over time. This phenomenon can be seen for all the time periods. As seen from Figure 5b,d, the same progression rates were found for both the 32-year and the 50-year cases. The 24-year case involved a similar situation.This is consistent with earlier forecasts. As seen from Figure 5, in the case of historic data, the $CO_2$ emission rate progressed after 1950 and this rate continued until the COVID-19 phenomenon observed in 2019. A similar upward progression will continue after the end of COVID-19 lockdowns. This progression rate might be lower only if the COVID-19 lockdown continues for a few more consecutive years. If that happens, then $CO_2$ emissions could be similar as shown in Figure 5b,d.

### 3.3. Effects Similar to COVID-19 Can Heal the Environment

As seen from the previous discussion, the COVID-19 effect can slow down the increasing progression of $CO_2$ emissions for a long time. In a low $CO_2$ emission atmosphere, there is much greater scope for the environment to heal its wounds. The most suitable case of the healing process is shown in Figure 5a,c, where three years (2020–2023) assuming the COVID-19 effect (lockdown) slow down the high progression rate of emissions. If this (lockdown or similar) can occur a number of times in a year, the progression rate can be lowered even more. A similar effect can be artificially induced by every nation so that $CO_2$ emissions decrease to a minimal level. Recently, Japan has been thinking of reducing its working days to 4 days a week. Microsoft Japan already implemented this program experimentally and found success [44].

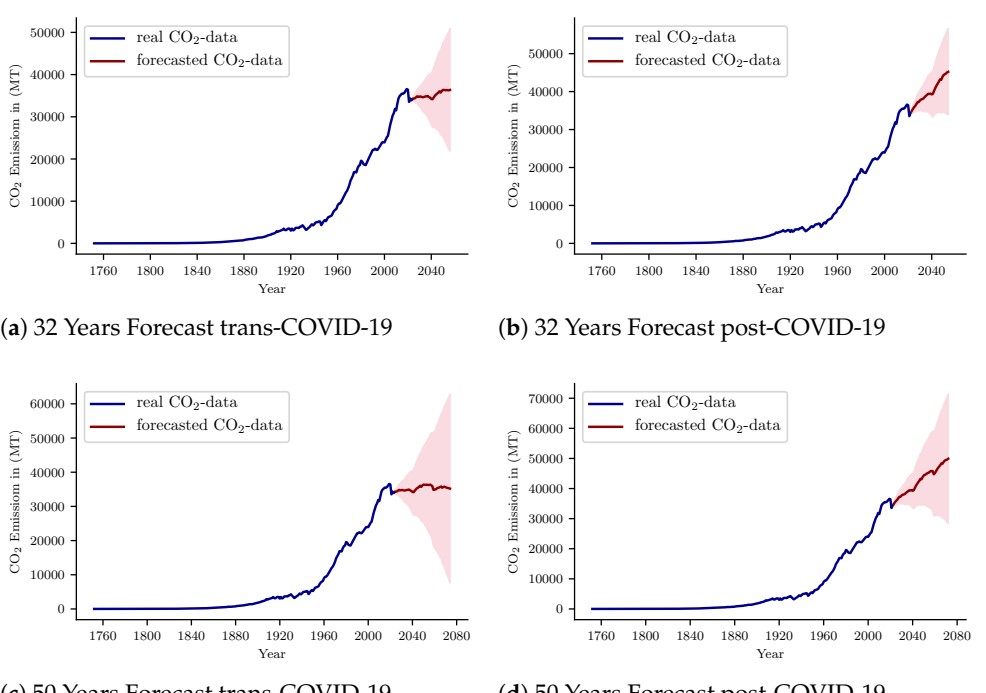

(**a**) 32 Years Forecast trans-COVID-19       (**b**) 32 Years Forecast post-COVID-19

(**c**) 50 Years Forecast trans-COVID-19       (**d**) 50 Years Forecast post-COVID-19

**Figure 5.** $CO_2$ Emission Near–Far Forecasting Behavior for 32 and 50 Years.

### 3.4. Consecutive CO_2 Emission Reduction Can Reduce Overall Emission Trend

Due to consecutive lockdowns across the world, a number of $CO_2$ emission sources have stopped emitting. As a result, the $CO_2$ footprint is lower than before. This phenomenon can be seen in Figure 5a,c. Here, only three years (2021, 2022, 2023) with reduced $CO_2$ emissions were involved. This little change in emissions has changed the overall emission pattern for a long time. With this result, a policy can be created to introduce artificial lockdown-like situations across nations to reduce $CO_2$ emissions to a viable point.

### 3.5. COVID-19 Helps Noticeable CO_2 Emission Reduction

As we have, COVID-19 has decreased $CO_2$ emissions to a significant extent. Here, we quantify the reduction footprint. As seen from Figure 6a, $CO_2$ emissions for the post-COVID-19 case (2022, 2023) are less than 3000 MT to 15,000 MT depending on the years addressed in the forecasting. A similar event can be observed for the transition course (2020, 2021) of COVID-19 (shown in blue). This different forecasting behavior is significant with regard to making a decision concerning developing policies with respect to which model to use for what kind of emission.

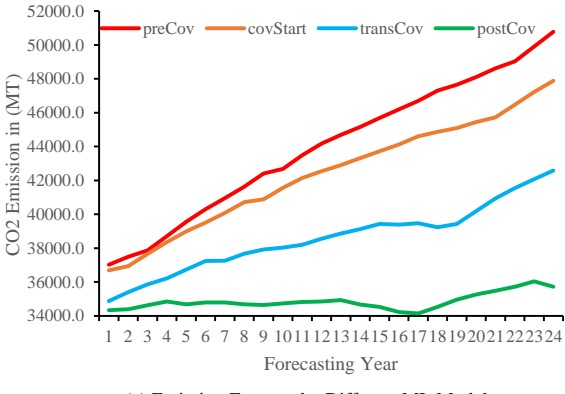

(a) Emission Forecast by Different ML Models

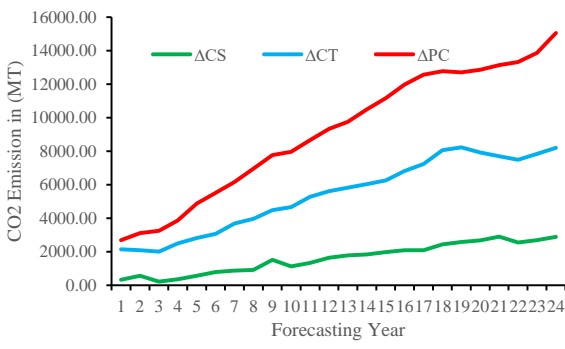

(b) Difference in Pre, Trans and Post COVID-19 Emission

**Figure 6.** $CO_2$ Emission Forecast Value Difference for Different ML Models.

*3.6. Long Lockdown Means Less $CO_2$ Emission*

An assumed lockdown for the case of the years 2020 to 2023 changed the forecasting value dramatically. These reduced values have many rational effects on the earth—the home of thousands of species. From Figure 6b, it is clearly visible that $\Delta PC$ is far bigger than $\Delta CS$ and $\Delta CT$ is in between the two. Here, $\Delta PC$, $\Delta CS$ and $\Delta CT$ are the forecasting value differences between post-COVID-19, start-COVID-19 and trans-COVID-19, respectively. As seen from $\Delta PC$ in Figure 6b, 3000 MT to 15,000 MT $CO_2$ less than the normal case $\Delta CS$ is emitted. This lesser emission of $CO_2$ can save a lot of resources across the world. This reduced $CO_2$ emission could be helpful concerning greenhouse effects, glacier ice melting, unexpected climate change, desertification, saltification of fresh water and soils near coastal areas and many more things which are harmful to the environment for the planet earth.

## 4. Discussion

All previous research presented in the literature had projected and modeled $CO_2$ emissions using data collected prior to the COVID-19 pandemic. Without accounting for the effects of COVID-19 on $CO_2$ emissions, predictions might include inaccuracies. This study accounted for COVID-19's effect on $CO_2$ emissions by including data from the previous declining trend. This yields realistic $CO_2$ emission predictions (up to 2000 MT reduction). Failing to account for COVID-19's effect on $CO_2$ emissions might make the prediction result more unrealistic (prediction difference is up to 15,000 MT). Mild prediction accuracy can also be observed for the COVID-19 transition period (prediction values up to 7000 MT). All the scenarios can be seen in Figure 6b. Hence, accounting for COVID-19's effect on $CO_2$ emissions yields realistic future $CO_2$ emission values.

On the other hand, the method used in this paper maps the predicted $CO_2$ emission values accurately. The maximum accuracy of model was 91%. Thus, the models developed are less error prone. Hence, the model-predicted $CO_2$ emission data and the accuracy

data converged. As a result, the method and predicted $CO_2$ emissions can be considered accurate and worth using. This claim is verified in Table 8.

**Table 8.** $CO_2$ Emissions (GT), Current vs. Forecast.

| Year | Current | Our Model |
|------|---------|-----------|
| 2022 | – | 36.45 |
| 2021 | 36.4 | 36.18 |
| 2020 | 34.8 | 35.20 |
| 2019 | 36.7 | 36.44 |

This paper is successful in terms of developing a robust and accurate $CO_2$ emission prediction model accounting for COVID-19-driven lockdown effects. It also delivered a number of meaningful and interesting insights from the historic data. The model is more accurate concerning $CO_2$ emission prediction than previous models. The model can be used to predict $CO_2$ emissions, to create policies for $CO_2$ emission reduction, and for $CO_2$ emission tracing. This modeling approach and the solution yielded can be considered new additions to the respective domains of knowledge.

In this paper, the authors focused on developing an optimized $CO_2$ emission prediction model. The work can be further extended by developing complete web or mobile applications to trace $CO_2$ emissions with the touch of finger tip. A comprehensive digital twin could also be developed for $CO_2$ emissions. All such works are options for future.

## 5. Conclusions

Carbon emissions, the greenhouse effect, climate change and catastrophic environmental issues have become the most crucial issues in the contemporary world. Application of AI and ML have a significant impact in terms of solving these issues. This work focuses on using AI to develop an ML model for global total $CO_2$ emissions to forecast $CO_2$ emissions for the near or far future. Building ML models considering reduced $CO_2$ emissions during the COVID-19 pandemic, we found some noticeable outcomes which can help in understanding $CO_2$ emissions across the world. The MAPEs for the four methods are: pre-COV (MAPE: 0.32), start-COV (MAPE: 0.28), trans-COV (MAPE: 0.19), post-COV (MAPE: 0.09), where the selected model to predict future $CO_2$ emission behavior has a MAPE of 9%. This is quite good accuracy with respect to the data available at hand. The post-COV model predicted global total $CO_2$ emissions for the years 2022 to 2027 are: 36,218.59, 36,733.69, 37,238.29, 37,260.88, 37,674.01 and 37,921.47 MT. By comparing our forecasting output to current and previous benchmark work, one can validate the obtained accuracy. Consequently, the forecasting of $CO_2$ emissions for the far future years should be accurate.

In this work, the post-COVID-19 model forecasts reasonable $CO_2$ emission behavior. Moreover, the trans-COVID-19 model shows some remarkable forecasts. Whatever the forecast we obtained, it may not actually reflect the real $CO_2$ emission practically. Moreover, a number of external influencing features need to be considered in future developments. Moreover, some other optimization methods or feature selections could be applied. All the remaining issues could be further explored in future research.

Further observations of this study are given below:

- The AI-based ML method can help to forecast $CO_2$ emission behavior during or after the COVID-19 pandemic.
- Lockdown-like situations can reduce $CO_2$ emissions in the present and in the far future.
- Artificial lockdowns or shorter (e.g., 4 days) working schedules can help to heal the environment.
- A policy can be created to impose artificial lockdown-like events to reduce the overall $CO_2$ emission footprint.

**Author Contributions:** Conceptualization, Y.M. and; methodology, Y.M.; software, Y.M.; validation, H.N.; formal analysis, Y.M.; investigation, Y.M.; resources, Y.M.; data curation, Y.M.; writing—original draft preparation, Y.M.; writing—review and editing, H.N.; visualization, Y.M.; supervision, Y.M.; project administration, Y.M.; funding acquisition, Y.M. All authors have read and agreed to the published version of the manuscript.

**Funding:** This research received no external funding.

**Institutional Review Board Statement:** Not applicable.

**Informed Consent Statement:** Not applicable.

**Data Availability Statement:** Data used in this article is cited in the reference section. More curated data is also available by a request to the corresponding author.

**Acknowledgments:** This work is owed to the assistant of Noman Hossain and his team for their cooperation and data acquisition role. It is worth mentioning some names those who are not written herein. This work is also grateful to the respective institutions for their instrumental support.

**Conflicts of Interest:** The authors declare no conflict of interest.

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
