# Peer review of "Predicting CO2 Emission Footprint Using AI through Machine Learning"

_atmosphere, doi:10.3390/atmos13111871_

Round 1
Author Response
Dear Reviewer,
We have revised our manuscript as per your suggestions.Please find the attachment.

Reviewer 2 Report
The present work could be reconsidered after addressing the following comments:
1. Add references for performance indicators (Eqs. 2-5)
2. Please avoid using the lumped citation like [8],[9],[10],[11],[12],[15].
3. Please add explanation about avoiding overfitting.
4. The manuscripts needs to cite recent works in this field.
5. Conclusion: The future scope of the work should be provided.
Author Response

(The authors gave the same response as above.)

Reviewer 3 Report
please see attached file

Author Response

(The authors gave the same response as above.)

Reviewer 4 Report
1. (1-6) An abstract is not a place to explain why CO2 emissions are important; rather, it is a place to describe what was done in the paper and what was not done.
2. (23) Why are reports like the IPCC not suitable for the COVID era? It seems to be a core thesis of the paper, yet no explanation of why this is the case is presented. Most important reports in this area—like the IPCC report (https://www.ipcc.ch/site/assets/uploads/2018/02/ar4-wg1-chapter10-1.pdf)—are not mentioned at all.
3. (29) Is there a controversy regarding that topic? It seems that there is a strong consensus on that matter, Cook, John, et al. "Consensus on consensus: a synthesis of consensus estimates on human-caused global warming." Environmental Research Letters 11.4 (2016): 048002.; Myers, Krista F., et al. "Consensus revisited: quantifying scientific agreement on climate changeand climate expertise among Earth scientists 10 years later." Environmental Research Letters 16.10 (2021): 104030.
4. (Figure 1) What is the source of the data that you use?
5. Description of SARIMA model (eq.(1) and below): no description of parameters B,φ,Θ,Φ,θ with top and bottom indexes;
6. (147-149) ARIMA (SURIMA?) model parameters are listed, but there is no S value (in seasonal ARIMA (SARIMA) models S value indicates seasonal length in the data); what seasonal length in the data was established?
7. (136) You mention using SARIMA, but in Table 2, SARIMA is not mentioned.
8. (Table 2) How were the values of the ARIMA model calculated? Why are these exact parameters selected? (lack of justification)
9. (Table 2) The CO2 estimate is a regression problem. What does an error of 0.32 exactly mean? What is the unit?
10. (Table 2) This is a regression problem, so what does "accuracy" mean exactly? Why does accuracy complement error to 1?
11. (Table 2) You use the method of taking only a few previous points and making the prediction for a year, yet you use it in Figure 4 to make a prediction for the next 50 years. Could you include the information about how big the error today is? We stop the data in the 1970s and try to make predictions of 50 years ahead? Would the error be still this small? Based on this error would you say that the prediction that you make for 2080 are accurate?
12. Has model validation been performed ?The out-of-time cross-validation can verify if model parameters are sensitive to time period.
13. The scores of prediction errors (MAPE,MSE, RMSE, MAD) are used to evaluate the model (eq.2-5); Why is no value given for any of these indicators, especially MAPE ?
14. (185) why the error of 0.32 means that "Hence, forecasting models are justified and accurate enough to represent real CO2 emission behaviour"?
15. (Figure 4) According to your figures, the estimated emissions of the lower and upper bounds of the estimation window vary by more than 200% (from around 25k to 50k) in 32 years and by 600% (from around 10k to 60k) in 50. How is this method any better than already existing predictions? How is basing the prediction ONLY on the assumption of a stationary process without taking into account any external factors, like the change in the prices of energy, regulation, or global demographics, be reliable? The method seems to be OK for predicting the emissions for a few years, but making predictions of anything further is an opinion not possible, with such a simple tool. The model used does not entitle forecasts for the next 32 years, much less 50 years.
16. (272-287) The conclusions are very vague and hardly refer to the results of the calculations carried out. Sentence:” developed models throw accuracy score of 81 to 91” This statement may be misleading. After all, it is impossible to say that the presented forecasts for 32 years or 50 years have such accuracy. Please specify to what time period the given model accuracy values were referred. Similarly conclusion: “Lockdown of 2 to 3 COVID-19 years can reduce CO2 emission for current time and the far future” is unauthorized. Only near future prediction is enough valid using SARIMA models.
Author Response

(The authors gave the same response as above.)

Round 2
Reviewer 1 Report
Review of "Predicting CO2 Emission Footprint using AI through Machine Learning".
Manuscript ID: atmosphere-1941752
Overall recommendation: Minor Revision
1- The authors must try to reduce the abstract section.
2- In Figure 2(c) and (d). Is it possible to enlarge the period of covid-19?
3- The authors consider this paper in the work explanation (http://dx.doi.org/10.3390/en12173263) but do not appear in the reference section.
4- There are numerous cases of minor grammatical inaccuracies and typos.
I hope these comments will be helpful to you.
Reviewer 2 Report
The authors have addressed my comments satisfactorily. I have no further questions.
Reviewer 3 Report
Please double check the writing to make sure all the grammar and spelling errors are corrected. Overall, the paper was well revised.
Reviewer 4 Report
Thank you for the reliable and comprehensive explanations and supplements to the paper.
I am satisfied with yours reply and I hope that the readers will appreciate the authors' contribution to the reliable preparation of an interesting case study.